# Antiviral efficacy of favipiravir against Zika and SARS-CoV-2 viruses in non-human primates

Romain Marlin [1,11], Delphine Desjardins [1,11], Vanessa Contreras[1,11], Guillaume Lingas[2,11], Caroline Solas[3,11], Pierre Roques [1,10,11], Thibaut Naninck [1], Quentin Pascal[1], Sylvie Behillil[4,5], Pauline Maisonnasse [1], Julien Lemaitre[1], Nidhal Kahlaoui[1], Benoit Delache[1], Andrés Pizzorno [6], Antoine Nougairede [7], Camille Ludot[1], Olivier Terrier [6], Nathalie Dereuddre-Bosquet [1], Francis Relouzat[1], Catherine Chapon[1], Raphael Ho Tsong Fang[1], Sylvie van der Werf [4,5], Manuel Rosa Calatrava[6,8], Denis Malvy[9], Xavier de Lamballerie [7], Jeremie Guedj [2,11] ✉ & Roger Le Grand [1,11] ✉

The COVID-19 pandemic has exemplified that rigorous evaluation in large animal models is key for translation from promising in vitro results to successful clinical implementation. Among the drugs that have been largely tested in clinical trials but failed so far to bring clear evidence of clinical efficacy is favipiravir, a nucleoside analogue with large spectrum activity against several RNA viruses in vitro and in small animal models. Here, we evaluate the antiviral activity of favipiravir against Zika or SARS-CoV-2 virus in cynomolgus macaques. In both models, high doses of favipiravir are initiated before infection and viral kinetics are evaluated during 7 to 15 days after infection. Favipiravir leads to a statistically significant reduction in plasma Zika viral load compared to untreated animals. However, favipiravir has no effects on SARS-CoV-2 viral kinetics, and 4 treated animals have to be euthanized due to rapid clinical deterioration, suggesting a potential role of favipiravir in disease worsening in SARS-CoV-2 infected animals. To summarize, favipiravir has an antiviral activity against Zika virus but not against SARS-CoV-2 infection in the cynomolgus macaque model. Our results support the clinical evaluation of favipiravir against Zika virus but they advocate against its use against SARS-CoV-2 infection.

Favipiravir (T-705) is an RNA polymerase inhibitor approved in Japan for the treatment of noncomplicated influenza infections and in clinical development in the United States. The drug has shown a strong antiviral activity against several RNA viruses in vitro[1] and in macaque models, including Ebola virus[2], Lassa virus[3], and Marburg[4] virus, making it an attractive candidate against emergent RNA viruses. During the 2013–2016 Ebola outbreak, the drug was evaluated in a clinical trial in Guinea[5], with no significant or definite effect on mortality, possibly due to suboptimal dosing regimens[6]. Few months later, favipiravir was also evaluated against Zika virus (ZIKV). It showed a strong antiviral activity in vitro[7], but its clinical impact could not be evaluated, due among others to the rapid decline of the epidemic. Results from mathematical modeling provided evidence that favipiravir could have a strong antiviral efficacy against ZIKV in cynomolgus macaques (CM)[7].

A list of author affiliations appears at the end of the paper. ✉e-mail: jeremie.guedj@inserm.fr; roger.le-grand@cea.fr

Favipiravir has also naturally been considered as a drug candidate against SARS-CoV-2. In vitro evaluation showed mixed results, with an antiviral activity of favipiravir, as measured by the 50% effective concentrations (EC50), ranging from 62 to >500 μM (10 to >78 μg/mL)[8–11]. The results were more encouraging in vivo, with favipiravir leading to reduction of infectious titers in lungs and clinical alleviation of the disease in the hamster model[12–14]. Given the complexity of the drug pharmacokinetics[15,16], an important finding of these studies was that the antiviral efficacy was achieved with plasma trough concentrations that were comparable or lower to those found during human clinical trials[6]. However toxicity signals were observed in some animals at the largest doses[12], and the translation to humans doses is made difficult by the rapid metabolic activity of rodents. Nonetheless, these results prompted a large interest due to the lack of *per os* antiviral drugs available, and favipiravir is currently being evaluated in more than 72 clinical trials registered in February 2022[17], both in ambulatory and hospitalized patients, making it the third largest evaluated antiviral drug administered to COVID-19 patients. Although some preliminary studies suggested that favipiravir could decrease the time to viral clearance in mild or moderate COVID-19 patients[18] or the time to clinical improvement[19], the retrospective aspect or the absence of randomization of most studies precludes solid conclusion on favipiravir efficacy.

In order to support ongoing and future clinical evaluations of favipiravir against SARS-CoV-2 and Zika infections, and more generally against future emerging RNA viruses, we designed three successive experiments in cynomolgus macaques (Fig. 1). We first provided a detailed description of favipiravir pharmacokinetic in uninfected animals over a 14 days repeated-dose experiments to define relevant dosing regimens. In a second experiment, we evaluated the antiviral efficacy of favipiravir in a CM model of Zika infection. In a third experiment, we evaluated the antiviral efficacy of favipiravir against SARS-CoV-2 infection in a CM model that reproduces human infection and makes possible the evaluation of drug efficacy in a well-controlled setting. We discuss the implications of our findings for favipiravir clinical evaluation against emerging or re-emerging RNA viruses.

## Results

### Selection of dosing regimen against Zika and SARS-CoV-2 virus infections

We first evaluated the pharmacokinetics of favipiravir in non-infected CM, using a loading dose of 250 mg/kg twice a day (BID) administered intravenously (IV), followed by repeated subcutaneous administrations of 150 mg/kg BID. In the four animals, the drug concentrations rapidly increased to achieve a median maximal ($C_{max}$) and trough ($C_{trough}$) concentrations of 309.1 and 75.2 μg/mL, respectively, after the loading dose (Fig. 2). Interestingly, favipiravir concentrations were maintained at high levels over the 14 days of the experiment, with trough concentrations of 79.1 and 131.6 μg/mL at day 7 and day 14, respectively (Fig. 2). These values are larger than the drug EC50 of favipiravir against ZIKV, that ranges between 2.2 and 6.6 μg/mL, supporting the choice of this dosing regimen in challenge experiments. In the case of SARS-CoV-2, the uncertainty in the exact value of EC50 (see "Discussion") led us to consider a larger spectrum of doses. Consistent with our previous studies[2], we used doses of 100/150/180 mg/kg BID to evaluate the full spectrum of efficacy and toxicity of favipiravir.

### Favipiravir pre-exposure prophylaxis reduces Zika virus replication

In a second experiment, we analyzed the antiviral efficacy of favipiravir against Zika virus. In this experiment, 6 control animals were untreated and infected by ZIKV (strain H/PF/2013), and 6 animals initiated favipiravir treatment 3 days before infection (pre-exposure prophylaxis), using the same dosing regimen than in the PK experiment (Fig. 1). Animals treated with favipiravir had lower levels of viral load than

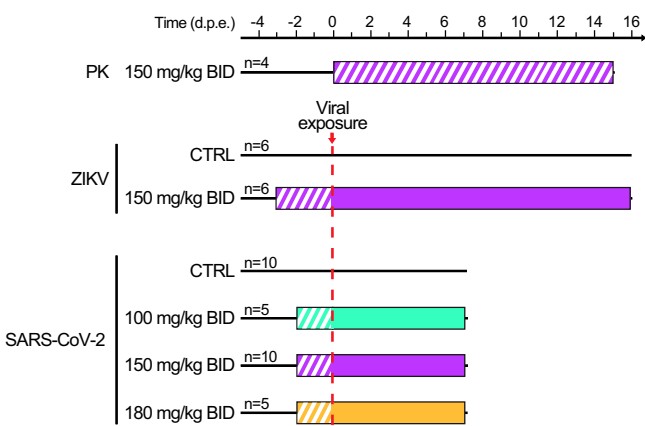

**Fig. 1 | Study design of the 3 experiments.** In the first experiment of pharmacokinetic (PK), $n = 4$ animals were treated with favipiravir (FPV) for 14 days. In the second experiment, $n = 12$ were either treated or received placebo, and were challenged with $10^6$ PFU of Zika virus (ZIKV) three days after treatment initiation. In the third experiment, $n = 30$ animals were either treated or received a placebo, and were challenged with $10^6$ PFU of SARS-CoV-2 two days after treatment initiation. Hatched area indicates FPV treatment without viral exposure. Colored areas indicate FPV dosing regimens; cyan: 200 mg/kg twice a day (BID) administered intravenously (i.v) on day −2 followed by 100 mg/kg BID administered subcutaneously (s.c); magenta: 250 mg/kg BID i.v. on day −3 (ZIKV) or −2 (SARS-CoV-2) followed by 150 mg/kg BID s.c.; yellow: 250 mg/kg BID i.v. on day −2 followed by 180 mg/kg BID sc. Untreated animals received NaCl 0.9% solution as placebo.

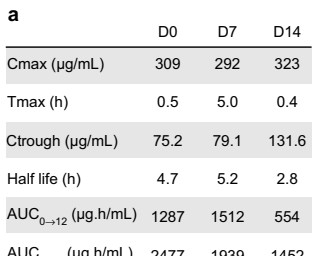
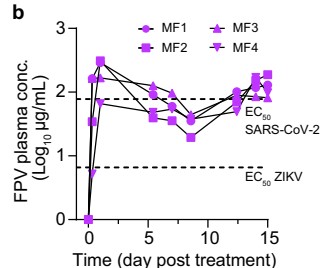

**a**

|  | D0 | D7 | D14 |
|---|---|---|---|
| Cmax (μg/mL) | 309 | 292 | 323 |
| Tmax (h) | 0.5 | 5.0 | 0.4 |
| Ctrough (μg/mL) | 75.2 | 79.1 | 131.6 |
| Half life (h) | 4.7 | 5.2 | 2.8 |
| AUC$_{0\rightarrow12}$ (μg.h/mL) | 1287 | 1512 | 554 |
| AUC$_{0\rightarrow\infty}$ (μg.h/mL) | 2477 | 1939 | 1452 |

**Fig. 2 | Plasma FPV concentration of four uninfected NHPs. a** Favipiravir pharmacokinetic parameters during treatment with 250 mg/kg BID i.v. on day 0 followed by 150 mg/kg BID s.c. for 14 days. **b** Longitudinal evolution of plasma FPV concentrations with respect to EC50 values obtained on ZIKV and SARS-CoV-2 given in refs. 7, 12. Source data are provided as a Source Data file.

untreated animals (Fig. 3a, b). Their viral kinetic profile was nonetheless less consistent than control animals, with some individuals experiencing longer duration of viral shedding. Overall, favipiravir had a significant effect on the peak viral replication, with median peak viral load of 5.6 and 6.5 log10 copies/mL in treated and untreated animals, respectively ($p = 0.026$, Fig. 3c). The overall viral shedding, as measured by the Area Under the Curve (AUC) from 0 to 7 days post exposure (dpe), was also significantly different between untreated and treated animals, with median values of 6.5 vs 5.9 log10 copies.day/mL in untreated and treated animals, respectively ($p = 0.041$, Fig. 3d). We also investigated a concentration-dependent effect of favipiravir, taking the geometric mean of plasma trough concentrations between 0 and 5 dpe as a surrogate of drug exposure (see "Methods"). Drug concentration showed a trend towards an effect on peak viral load and on AUC viral load ($p = 0.056$ and $p = 0.074$, respectively, Fig. 3e, f), suggesting that high concentrations could be associated with a reduction of viral load, with a nonlinear relationship.

Beside drug concentrations, several treated animals exhibited distinctive cytokine dynamics compared to control animals (Figs. 3g and S2). Indeed, levels of IL-1RA and CCL2 peaked at 1 dpe and went

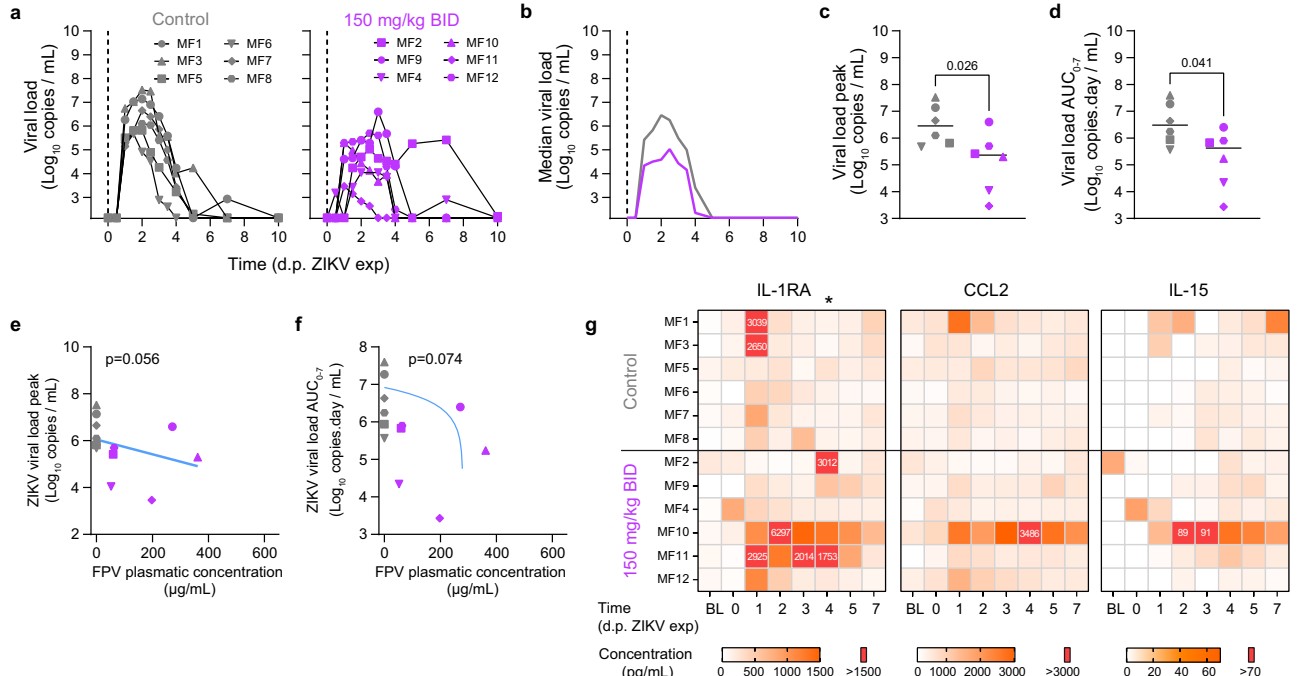

**Fig. 3 | Viral kinetic and pharmacokinetic in ZIKV infected cynomolgus macaques treated with FPV. a** Individual plasma viral loads determined by RT-PCR in all animals; **b** Median plasma viral load values observed in each treatment group; **c, d** Viral kinetic parameters (peak and AUC viral load) during the first 7 days of infection. Median value is indicated by horizontal bar. Parameters were compared between groups using the two-tailed non-parametric Mann–Whitney test. **e, f** Viral kinetic parameters (peak and AUC viral load between 0 and 7 dpi) according to geometric mean FPV plasma trough concentration. Grey: untreated; Purple: 150 mg/kg BID. A Spearman correlation test was performed to assess the association between drug concentration and viral kinetic parameters. Two-tailed $p$ values are indicated. **g** Heatmaps of the concentrations of IL-1RA, CCL2, and IL-15 measured in plasma ZIKV infected animals. The asterisk indicates a significant difference in the concentration of IL-1RA at 4 d.p.e. between the control group and the FPV group. Parameters were compared between groups using the two-tailed non-parametric Mann–Whitney test. The color scale (in pg.mL$^{-1}$) is shown at the bottom. Source data are provided as a Source Data file.

back to basal level at 2/3 dpe for most animals, whereas the pro-inflammatory factors were sustained at high concentration until 5-7 dpe for treated animals, suggesting that they could be causally related to favipiravir administration (Fig. 3g).

Overall the treatment was well tolerated, even if the median loss weight at day 7 post treatment initiation was more elevated in treated (6.71%) than control (2.96%) animals ($p = 0.093$, Figs. S1b and S1d). Biochemistry parameters tended to be impacted in treated animals, especially with alteration of liver function and metabolism. In fact, the treated animals, which exhibited sustained inflammation, showed also markers of hepatic cytolysis (ASAT elevation), slight cholestasis (GGT elevation) associated with increase uremia and lipidaemia after ZIKV exposure, whereas these parameters remained unchanged in untreated animals (Fig. S3).

### Favipiravir pre-exposure prophylaxis does not reduce SARS-CoV-2 virus replication

Following what was done for hydroxychloroquine[20], we studied in vitro antiviral activity of favipiravir against SARS-CoV-2 infection in the reconstituted human airway epithelium MucilAir™ model (HAE). Doses of 200–600 µM failing to reduce significantly SARS-CoV-2 apical viral titers at 48 h post infection, and did not protect the epithelial integrity during infection (Fig. S4).

We next tested the efficacy of favipiravir in the CM model of SARS-CoV-2 infection following exposure to $1 \times 10^6$ pfu of SARS-CoV-2 (hCoV-19/France/IDF0372/2020) by combined nasopharyngeal and tracheal routes as we previously reported[20–22] (Fig. S10). Ten control animals were left untreated and infected by SARS-CoV-2 virus, and 20 animals initiated favipiravir treatment 2 days before infection (pre-exposure prophylaxis), with a loading dose followed by maintenance doses of 100 ($n = 5$), 150 ($n = 10$) and 180 ($n = 5$) mg/kg BID. Viral kinetics in

nasopharyngeal compartment were similar between untreated and treated animals, irrespective of the dose (Fig. 4a, b). Peak viral load in nasal fluid was largely similar in all groups, with median values of 8.2, 7.2, 8.3, and 8.1 log$_{10}$ copies/mL in untreated, 100 mg, 150 mg, and 180 mg/kg BID groups respectively (Fig. 4c). Similar results were obtained for the AUC viral load in the nasal fluid, with median values of 8.5, 7.4, 8.6, and 8.4 log$_{10}$ copies day/mL in untreated, 100 mg, 150 mg and 180 mg/kg BID groups, respectively (Fig. 4d). Results were largely similar in the tracheal fluid (Fig. S5), although a larger peak viral load was observed in animals treated with 180 mg/kg BID as compared to untreated (median values of 8.0 and 7.4 log$_{10}$ copies/mL, respectively, $p = 0.003$, Fig. S5c). There was no effect of drug concentration on nasopharyngeal peak viral load ($p = 0.64$ Fig. 4f), but there was a trend towards an effect of favipiravir concentration in increasing AUC viral load ($p = 0.084$, Fig. 4g). The same results were observed in tracheal fluid, with a trend towards tracheal peak viral loads ($p = 0.087$) and AUC viral load (Supplementary Fig. S5e, f).

Viral load was also measured in BAL at 3 dpe (Fig. 4e). There was a trend towards larger viral loads in treated animals, in particular at the larger doses. While control animals had a median peak viral load at 6.39 log$_{10}$ copies/mL, these values were equal to 6.32, 7.08, and 7.80 log$_{10}$ copies/mL in treated animals at the dose of 100, 150, and 180 mg/kg BID, respectively ($p$-value to controls of 0.768, 0.143, and 0.005, respectively). The favipiravir concentrations determined in the epithelial lining fluid (ELF) were also correlated with the BAL viral load ($p < 0.0001$, Fig. 4h). Similar trends were observed in lung tissues at euthanasia (Fig. S5g, h).

Untreated infected animals exhibited mild clinical signs, consistent with a disease being often asymptomatic or mildly symptomatic in humans, with coughing or sneezing without dyspnea in SARS-CoV-2 infected animals (Fig. S6). Slight weight loss was observed in all

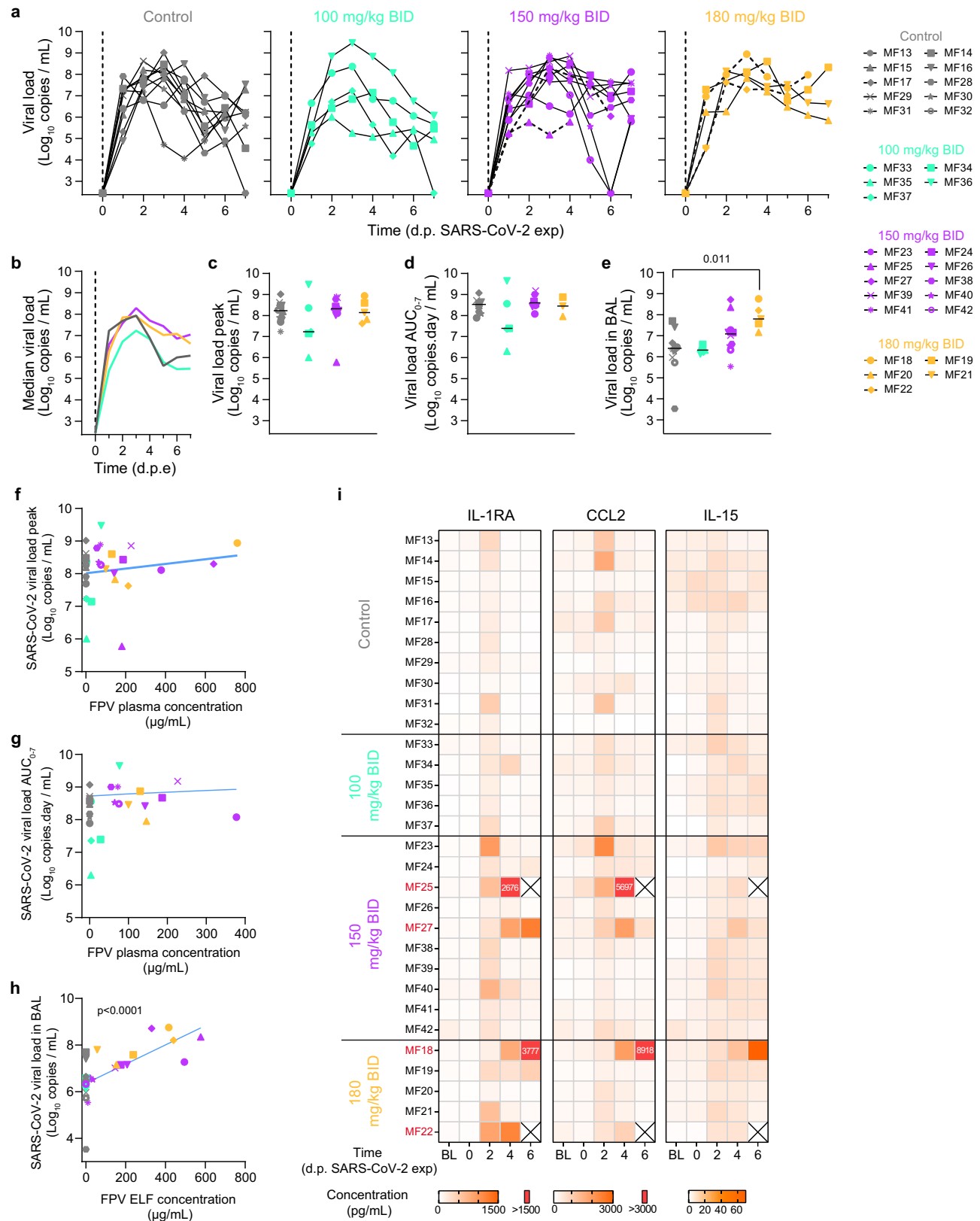

untreated infected animals that could be due to either infection and/or repeated anesthesia (Fig. S1c). Treatment led to weight loss, with a median weight loss after 7 days of treatment equal to 2.78, 6.23, 4.87, and 6.44% in untreated, 100 mg, 150 mg, and 180 mg/kg BID groups, respectively (*p*-value to controls of 0.001, 0.006, and 0.055, respectively) (Fig. S1d).

## Exacerbation of SARS-CoV-2 disease in favipiravir treated macaques

Similar to previous observations[20,23] animals infected by SARS-CoV-2 showed transient elevated levels of IL-1RA, CCL2, and IL15, that peaked at 2 dpe. As observed during ZIKV infection, levels of these pro-inflammatory factors were sustained or continued to increase after 2

**Fig. 4 | Viral kinetic and pharmacokinetic in the respiratory tract of SARS-CoV-2-infected cynomolgus macaques treated with FPV. a** Individual nasopharyngeal viral loads determined by RT-PCR in all animals; **b** Median nasopharyngeal viral load values observed in each treatment group; **c**, **d** Viral kinetic parameters (peak and AUC viral load) during the first 7 days of infection. **e** Viral load in bronchoalveolar lavages (BAL) at 3 dpe. Median value is indicated by horizontal bar. Parameters were compared between groups using Kruskal-Wallis test following Dunn's multiple comparisons. **f**, **g** Viral kinetic parameters (peak and AUC viral load between 0 and 7 dpi) according to geometric mean FPV plasma trough concentration. **h** Viral load in BAL according to FPV concentration in the epithelial lining fluid (ELF). A Spearman correlation test was performed to assess the association between drug concentration and viral kinetic parameters. Two-tailed $p$ value is indicated. Grey: untreated; cyan: 100 mg/kg BID; purple: 150 mg/kg BID; yellow: 180 mg/kg BID. **i** Heatmaps of the concentrations of IL-1RA, CCL2 and IL-15 measured in plasma SARS-CoV-2 infected animals. The color scale (in $pg.mL^{-1}$) is shown at the bottom. Source data are provided as a Source Data file.

dpe in treated animals, suggesting that they could be exacerbated by the administration of favipiravir. Consistent with this interpretation, there was a dose-dependent effect on cytokines, with larger levels observed in SARS-CoV-2 infected animals treated with 150 or 180 mg/kg BID than those untreated or treated with 100 mg/kg BID (Fig. 4i). In treated animals, 4/20 animals (MF18, MF22, MF25, and MF27) had to be euthanized due to rapid deterioration of their clinical score, and all these animals were infected with SARS-CoV-2 (2 receiving 150 mg/kg BID and 2 receiving 180 mg/kg BID). In these 4 animals, the levels of IL-1RA and CCL2 remained large at all times, and similar observations could be made on other cytokines (Fig. S7). Veterinary examination showed a shock in these four animals with severe hypothermia, bradycardia, with or without electrocardiographic abnormalities, hypoxemia, discordance, tachypnea, hypotension neutrophilia, and lymphopenia. Blood chemistry showed an increase of transaminases (ASAT and ALAT) without change in PAL or GGT suggesting hepatocellular necrosis (Fig. 5a and Fig. S8). In these animals, metabolism impairment was observed with an increase of plasma triglycerides and decrease of fructosamine and cholesterol levels with a similar kinetic than clinical score (Figs. S8 and S6). This interpretation is also corroborated by an increased levels of plasma creatinine in two animals from high dose group suggesting acute kidney failure. Moreover, metabolism alteration was confirmed by the continued increase of favipiravir plasma concentration, the reduction of M1 metabolite/favipiravir plasma ratio over time (Fig. 5b) and an accumulation of favipiravir in tissues (Fig. S9a). Ultrasounds and CT imaging performed 5 dpe and histological analyses showed typical feature of severe form of SARS-CoV-2 infection such as acute interstitial pneumonia with significant pleural effusion and left ventricle dilatation along with hepatomegaly with severe liver steatosis (Fig. 5c–e and Fig. S9c). Lesions on tissues were also noticed on other animals at a lower extent (Fig. S9b). Furthermore, high amount of virus was found in lung tissues of these four animals (Fig. 5f and Fig. S9d). Altogether, this suggest that animals had a multiple organs dysfunction syndrome (MODS) with liver and cardiac failure, associated with an acute kidney failure in two animals. This is consistent with the elevated levels of CCL2 and IL-1RA observed in all four animals, showing an immune activation which part of MODS physiopathology.

## Discussion

Here we showed the results of experiments in cynomolgus macaques to evaluate the antiviral effect of favipiravir against Zika and SARS-CoV-2 viruses. We determined dosing regimens that achieved relevant drug concentrations with respect to drug $EC_{50}$ and we evaluated the effects of pre-exposure favipiravir treatment on viral kinetics. Favipiravir significantly reduced Zika viral replication while being well tolerated at 150 mg/mL BID. However, it had no antiviral effect against SARS-CoV-2 at any of the doses tested (ranging 100–180 mg/kg BID), and 4/20 treated animals had a rapid clinical deterioration which required premature euthanasia during the study.

The effects of favipiravir on Zika virus viremia in macaques unambiguously demonstrate for the first time in vivo antiviral activity of favipiravir. As mentioned for other viral diseases, favipiravir can penetrate the sexual compartments and cross the blood brain barrier, facilitating the purge of the reservoirs of the virus[24]. In addition to mono-therapy, favipiravir may represent a good candidate for combined therapies with antiviral drugs such as galidesivir, which was also reported highly active in the NHP model[25]. Although the probable teratogenicity of favipiravir prevents its use in pregnant women, its ease of storage and its oral administration make it particularly relevant as a first line of protection to administer to suspect or contact cases.

The results obtained against SARS-CoV-2 infection are in contradiction with those in the hamster model[12–14]. This may stem from several factors, that are not mutually exclusive. First, the drug pharmacokinetics in NHPs differ from that in hamsters that was characterized by rapid drug metabolism, reducing its toxic effects and offering the possibility to administer large doses in hamsters[26]. For instance, we estimated previously that the effective dose to reduce viral replication by 90% ($ED_{90}$) in hamster in preventive therapy was 35 mg/day, which corresponds to about 600 mg/kg BID. Using the classical allometric rules, this would correspond to a dose of 200 mg/kg BID in NHPs, which is higher than the 100–180 mg/kg BID range tested here. However, the results found in HAE system, with no viral activity observed at doses up to 600 μM (Fig. S4) suggest that concentrations may need to be very high to generate a strong antiviral effect. In fact, our results even suggested an enhanced viral replication effect, with larger viral load levels in highest dosing group regimens than in untreated animals, despite mean plasma favipiravir concentrations well above the highest $EC_{50}$ value and good diffusion into lung tissue (mean lung/plasma ratio 86%). Whether this is a genuine effect or is simply coincidental is unknown, but it is noteworthy that similar results were also reported in previous ex vivo models[27]. Regardless of the origin of this difference between the hamster and the NHP models, the dosing regimens used here in NHPs correspond to doses in humans that are already larger than 1200 mg BID (see refs. 2, 24 for a detailed analysis of the correspondence between NHP and human pharmacokinetics). This is to be compared with the dose of 600–800 mg BID favipiravir usually administered, and is consistent with the lack of antiviral activity that has been reported in randomized clinical trials[28,29]. In fact our results show that even larger doses are unlikely to generate an antiviral activity, and therefore do not support the use of favipiravir against SARS-CoV-2 infection.

SARS-CoV-2 infection results in several extrapulmonary manifestations, including kidney and liver injury[30]. Angiotensin-converting enzyme 2 (ACE2), the entry receptor for SARS-CoV-2, is expressed in the liver and kidney[31–33] but viral replication in these organs remains controversial[34,35]. Both kidney and liver injury are likely multifactorial involving direct effects of the virus with inflammation and tissue damages, but also indirect effects resulting from systemic inflammation, dysregulated immune responses, endothelial dysfunction, and impaired organ crosstalk[32,35,36]. Elevation of liver enzymes have been reported during SARS-CoV-2 infection, however, liver injury may be a reflection of a severe form of the disease[32]. Hepatic injury associated with COVID-19 seems due to systemic inflammation and multi-organ dysfunction[36]. Our results showed also an exacerbation of SARS-CoV-2 disease in four animals treated with favipiravir, with an impact on liver and kidney functions. Importantly this effect had not been seen in previous experiments involving favipiravir and repeated anesthesia nor in infected untreated animals[15], suggesting that this effect was causally related to the dual effects of favipiravir administration and SARS-CoV-2 infection. Effects on liver functions and liver enzyme levels were reported as main adverse effects in COVID-19 patients

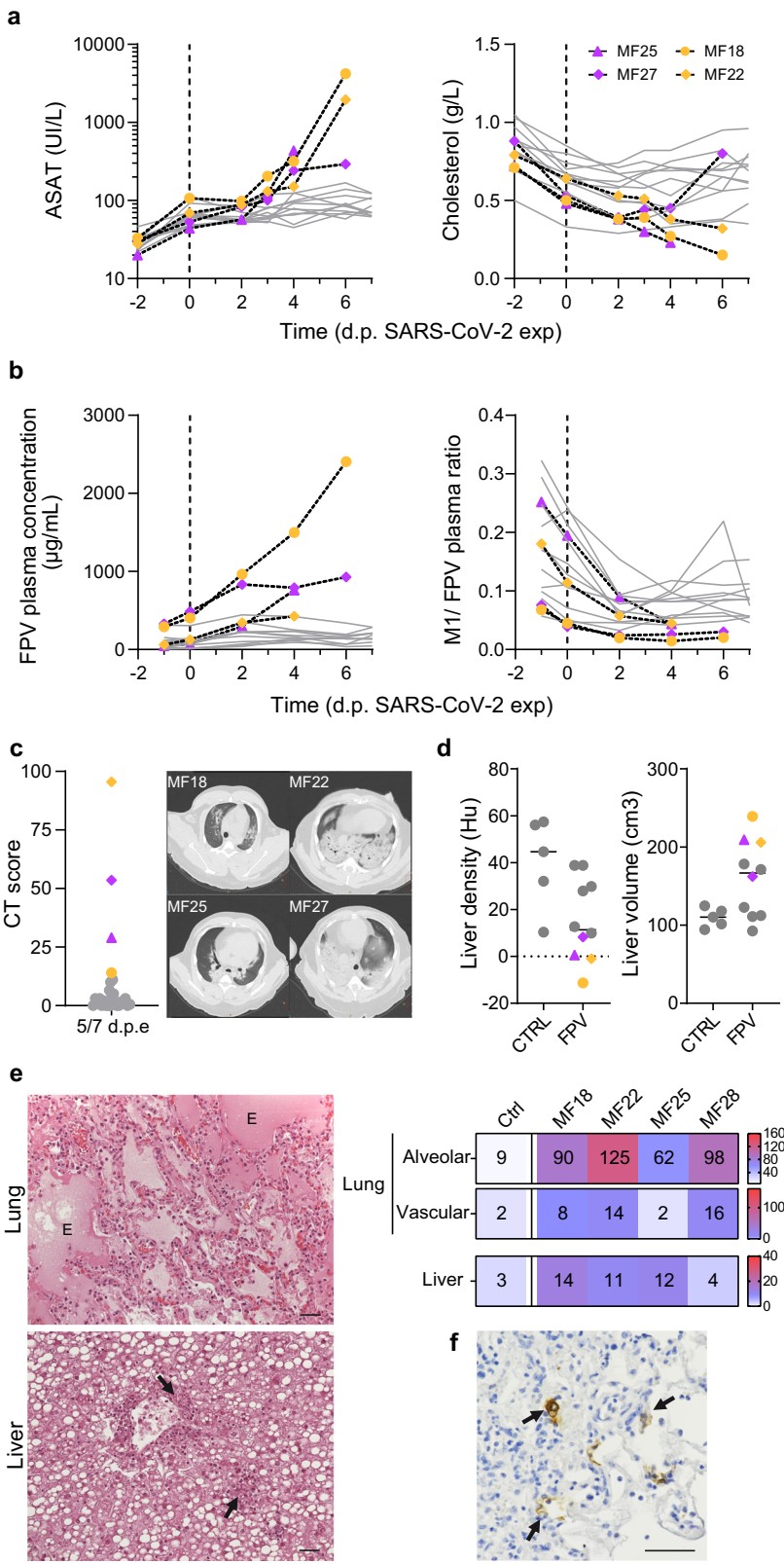

that received favipiravir treatment[29]. Favipiravir metabolization by the liver may increase the susceptibility to liver injury with SARS-CoV-2 infection and thus revealed liver pathogenicity.

To summarize, our results show that favipiravir has an antiviral activity against Zika virus but not against SARS-CoV-2 infection in the cynomolgus macaque model. They do not support the use of favipiravir in humans against SARS-CoV-2 infection.

## Methods

### Ethics and biosafety statement

Cynomolgus macaques (*Macaca fascicularis*), aged 44-86 months (20 females and 22 males, Table S1) and originating from Mauritian AAALAC certified breeding centers were used in this study. All animals were housed in IDMIT facilities (CEA, Fontenay-aux-roses), under BSL-2 and BSL-3 containment when necessary (Animal facility authorization #D92-

**Fig. 5 | Exacerbation of SARS-CoV-2 induced disease in four FPV-treated macaques.** Four infected animals treated with FPV were early euthanized after reach of humane endpoint. **a** Analyze of ASAT and Cholesterol levels in plasma of treated infected animals. **b** Longitudinal evolution of FPV concentration and M1/FPV ratio in plasma. **c** Lung lesions were assessed by chest CT at 5 dpe. Overall CT score are indicated, historical untreated animal were showed in grey. Representative images of lung lesions in the four NHPs. **d** Liver density and volume was assessed by CT scan at 5 dpe. Values for MF18, MF22, MF25, and MF27 are indicated in color according to the FPV dose (pink: 150 mg/kg BID and yellow: 180 mg/kg

BID), and other FPV treated animals were indicated in grey. Median value is indicated by horizontal bar. Source data are provided as a Source Data file. **e** Tissue lesions and cell infiltrates were analyzed at necropsy and histological score for lung (alveolar and vascular areas) and liver were shown for the 4 NHPs in comparison with control animals (median of $n = 5$). Representative images of lung (top) and liver (down) were shown. The letter E indicates presence of edema and black arrows show the neutrophilic infiltration. **f** Presence of SARS-CoV-2 infected cells in lung tissue was exhibited anti-Nucleocapsid antibody. Black arrows indicated infected cells. The black bars indicate 50 µm.

032-02, Prefecture des Hauts de Seine, France) and in compliance with European Directive 2010/63/EU, the French regulations and the Standards for Human Care and Use of Laboratory Animals, of the Office for Laboratory Animal Welfare (OLAW, assurance number #A5826-01, US). The protocols were approved by the institutional ethical committee "Comité d'Ethique en Expérimentation Animale du Commissariat à l'Energie Atomique et aux Energies Alternatives" (CEtEA #44) under statement numbers A16-013 and A20-011. These studies were authorized by the "Research, Innovation and Education Ministry" under registration numbers APAFIS#4079-2016021212132792v3 and APAFIS#24434-2020030216532863v1.

### Design of the experiments
In the first experiment, the pharmacokinetics of FPV was assessed in 4 uninfected animals, that received a loading dose (250 mg/kg) BID by intravenous route on the first day and a maintenance dose (150 mg/kg) BID for 14 days by subcutaneous route (simply called "150 mg/kg BID" in the following).

In the second experiment, 12 animals (including the four animals from the first experiment) were randomly assigned to the same dosing regimen or received NaCl 0.9% solution as control (simply called "untreated" in the following). Three days after treatment initiation, animals were exposed to $10^6$ pfu of H/PF/13 Zika strain via subcutaneous route. All animals were followed for at least 14 days post exposure. Blood sampling was performed all along the study to quantify circulating levels of ZIKV, to determine the concentrations of favipiravir and cytokines in plasma. Animals were euthanized between 14 and 16 dpe.

In the third experiment, favipiravir was evaluated for the treatment of SARS-CoV-2. In our animal model, SARS-CoV-2 infection is similar in male and female cynomolgus macaques (Fig. S10). Animals were randomized to the same dosing regimen or were left untreated, as well as a larger maintenance dose of 180 mg/kg BID (called "180 mg/kg BID" in the following) or a lower dosing regimen group with a loading dose of 200 mg/kg BID on day 0 followed by a maintenance dose of 100 mg/kg BID (called "100 mg/kg BID" in the following). Two days after treatment initiation, all animals were exposed to $10^6$ pfu of SARS-CoV-2 (hCoV-19/France/IDF0372/2020 strain; GISAID EpiCoV platform under accession number EPI_ISL_406596) via the combination of intranasal and intra-tracheal routes (Day 0), using atropine (0.04 mg/kg) for pre-medication and ketamine (5 mg/kg) with medetomidine (0.05 mg/kg) for anesthesia. All animals were euthanized at day 7 post exposure.

In all experiments, animals were observed daily and clinical exams were performed at baseline, daily on anaesthetized animals using ketamine (5 mg/kg) and medetomidine (0.05 mg/kg). During SARS-CoV-2 infection follow-up, body weight, rectal temperature, food/water consumption, activity, dehydration, respiration, heart rates, and oxygen saturation were recorded in a scoring grid. If animals reached the humane end point score, euthanasia was performed. Blood, as well as nasopharyngeal, tracheal, and rectal swabs, were collected among time. Broncho-alveolar lavages (BAL) were performed using 50 mL sterile saline on 3 and 6 dpi. Chest CT was performed on 5 dpi in anesthetized animals using tiletamine (5 mg/kg) and zolazepam (5 mg/kg). Blood cell counts, haemoglobin and haematocrit were determined

from EDTA blood using a DXH800 analyzer (Beckman Coulter). Biochemistry parameters were analyzed with standard kits (Siemens) and with a canine kit (Randox) in lithium heparin plasma, inactivated with Triton X-100, using ADVIA1800 analyzer (Siemens). Cytokines were quantified in EDTA-treated plasma using NHP Milliplex (Millipore) and a Bioplex 200 analyser (Bio-Rad) according to the manufacturer's instructions.

### Virus quantification in cynomolgus macaque samples
Plasma samples were collected from EDTA blood. Analysis by RT-qPCR using primers and probes derived from[37] (ZIKV_F, and ZIKV_R) encompassing a small segment coding for the E protein. Briefly, RNA was purified from 100 µL of plasma, using the Nucleospin 96 Virus Kit (Macherey Nagel, Düren, Germany, ref: 740452.4) according to the manufacturer's instructions. RNA was eluted in 100 µL of nuclease-free water and stored at −80 °C until analysis. ZIKV viral stock, diluted in an EDTA-plasma sample from ZIKV-non-infected macaques was used to generate a standard curve by serial 10-fold dilutions. Three aliquots of the ZIKV stock and two EDTA-plasma samples from ZIKV-negative macaques were used as positive and negative RT-qPCR controls, respectively. Then, 10 µL of the extracted RNA was mixed with the QRT-PCR medium containing primers, probes, enzyme and buffer (Supersript III platinum one step qPCR system from Invitrogen, Villebon-sur-Yvette, France) in a 96 well plate and ran on a Bio-Rad CFX thermocycler (Bio-Rad Lab., Marnes-la-Coquette, France). Results were quantified relative to ZIKV Vero supernatant diluted from $10^7$ copies/mL to 330 copies/mL that was previously calibrated as described in ref. 38. Lower limit of quantification (LOQ) = 2.70 $\log_{10}$ copies of ZIKV RNA per mL; Lower limit of detection (LOD) = 2 $\log_{10}$ copies/mL. Data were analysed with CFX Maestro (V2.2)

Upper respiratory (nasopharyngeal and tracheal) and rectal specimens were collected with swabs (Viral Transport Medium, CDC, DSR-052-01). Tracheal swabs were performed by insertion of the swab above the tip of the epiglottis into the upper trachea at approximately 1.5 cm of the epiglottis. All specimens were stored between 2 and 8 °C until analysis by RT-qPCR with a plasmid standard concentration range containing an *RdRp* gene fragment including the *RdRp-IP4* RT-PCR target sequence (Supplementary Table II). The limit of detection was estimated to be 2.67 $\log_{10}$ copies of SARS-CoV-2 gRNA per mL and the limit of quantification was estimated to be 3.67 $\log_{10}$ copies/mL. The protocol describing the procedure for the detection of SARS-CoV-2 is available on the WHO website (https://www.who.int/docs/default-source/coronavirus/real-time-rt-pcr-assays-for-the-detection-of-sars-cov-2-institut-pasteur-paris.pdf?sfvrsn=3662fcb6_2).

### CT scan imaging of SARS-CoV-2 infected animals
CT acquisitions were performed under breath-hold using the Digital Photon Counting (DPC) PET-CT system (Vereos-Ingenuity, Philips) implemented in a BSL-3 laboratory. The CT detector collimation used was 64 × 0.6 mm, the tube voltage was 120 kV, and the intensity was approximately 150 mAs. The intensity was regulated by automatic dose optimization tools. CT images were reconstructed with a slice thickness of 1.25 mm and an interval of 0.63 mm. Images were analyzed using INTELLISPACE PORTAL (V8, Philips Healthcare) and 3DSlicer (open-source tool; Version 5). All lung images had the same window

level of −300 and a window width of 1600. Pulmonary lesions were defined as ground-glass opacity, crazy-paving pattern, or consolidations, as previously described[20–23,39]. Two to three individuals assessed the lesion features detected by CT imaging independently and the final CT score results were determined by consensus. Pre-existing background lesions or lesions induced by experimental atelectasis were scored 0. Liver volume and average density were assessed following organ segmentation in 3DSlicer software.

## Pharmacokinetic assessment

Quantification of favipiravir, in plasma, BAL and tissues was performed by a validated sensitive and selective validated high-performance liquid chromatography coupled with tandem mass spectrometry (HPLC-MS/MS) method (UPLC-TQD, Waters, USA) with a lower limit of quantification of 0.1 μg/mL as previously described (12). Its major circulating metabolite M1 was quantified in plasma and tissues using to the same validated LC-MS/MS method. Lung biopsies collected after euthanasia were thoroughly rinsed with cold 0.9% NaCl to remove blood contamination and blotted with filter paper. Then, each lung biopsy was weighed and homogenized with 1 ml of 0.9% NaCl using a Mixer mill MM200 (Retsch, Germany). Cellular debris was removed by centrifugation, and the supernatant was stored at −80 °C. Favipiravir and M1 were extracted by a simple protein precipitation method, using acetonitrile for plasma and ice-cold acetonitrile for clarified tissue homogenates. Briefly, 50 μL of samples matrix was added to 500 μL of acetonitrile solution containing the internal standard (favipiravir-13C,15N, Alsachim), then vortexed for 2 min followed by centrifugation for 10 min at 4 °C. The supernatant medium was evaporated and the dry residues were then transferred to 96-well plates and 50 μL was injected.

Drug accumulation in lung was assessed by calculating a tissue to plasma concentration ratio. The favipiravir concentration in the epithelial lining fluid (ELF) were obtained from measured BAL fluid concentrations ($C_{BAL}$) after correction using the urea dilution method: $C_{ELF} = C_{BAL} \times (Urea_{plasma}/Urea_{BAL})$, where $Urea_{BAL}$ and $Urea_{plasma}$ correspond to the concentrations of urea determined in BAL fluid and plasma, respectively. Urea was determined in BAL by LC-MS/MS with a limit of quantification of 1.25 μg/mL as previously described[40].

## Favipiravir pharmacokinetics and effects on viral kinetic parameters

The following pharmacokinetic parameters of plasma favipiravir were calculated in the 4 treated uninfected animals through non-compartmental analysis (NCA) and are presented as median: $C_{max}$, $T_{max}$, half-life, $C_{min}$, areas under curve (AUC): $AUC_{0\to12}$ and $AUC_{0\to\infty}$. $C_{min}$ are the concentrations extrapolated 10 h after treatment administration using $\lambda_z$ regression determined with PkAnalix.

The following viral kinetic parameters were calculated in all infected animals: peak viral load, logarithm of the AUC of viral load between 0 and 7 dpe. In treated infected animals, we used the geometric mean concentrations of pre-dose favipiravir observed between 0 and 5 dpe (ZIKV), and 0 and 7 dpe (SARS-CoV-2) as a pharmacokinetic parameter reflecting the exposure to favipiravir.

## In vitro efficacy of favipiravir against SARS-CoV-2 infection in the HAE system

MucilAir™ HAE reconstituted from human primary cells obtained from bronchial biopsies, were provided by Epithelix SARL (Ref. EP01MD, Geneva, Switzerland) and maintained in air–liquid interphase with specific culture medium in Costar Transwell inserts (Corning, NY, USA) according to the manufacturer's instructions. For infection experiments, apical poles were infected with a 150 μl dilution of virus in OptiMEM medium (Gibco, ThermoFisher Scientific), at a multiplicity of infection (MOI) of 0.1. Treatments with FPV were applied through basolateral poles. FPV treatments were performed on days -2, -1, 0 (1 h

after viral infection), and 1 post infection. Samples were collected at 48 hpi. Variations in transepithelial electrical resistance (Δ TEER) were measured using a dedicated volt-ohm meter (EVOM2, Epithelial Volt/Ohm Meter for TEER) and expressed as Ohm.cm2.

**Statistical analysis.** Data were collected using classical Excel files (Microsoft Excel 2016). Differences between unmatched groups were compared using the Mann–Whitney U test and Kruskal–Wallis test following Dunn's multiple comparisons (Graphpad Prism 8.0). The following viral kinetic parameters were calculated in each experimental group as medians: viral load peak, logarithm of the area under the curve of the viral load. To evaluate a potential effect of drug exposure on viral dynamics, we further evaluated the correlation of the viral kinetic parameters with the plasma concentrations of FPV, taking the geometric mean trough concentrations observed in each infected macaque between 0 and 7 days after infection as a marker of drug exposure (Spearman test).

## Reporting summary

Further information on research design is available in the Nature Research Reporting Summary linked to this article.

## Data availability

Source data are provided with this paper. The data generated in this study are provided in the Source Data file. Source data are provided with this paper.

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

## Acknowledgements

We thank Sebastien Langlois, Quentin Sconosciuti, Victor Magneron, Claire-Maelle Fovet, Johana Demilly, Nina Dhooge, Pauline Le Calvez, Maxime Potier, Jen-Marie Robert, Orianne Lacroix, Christophe Joubert, Thierry Prot and Cristina Dodan for the NHP experiments; Laetitia Bos-sevot, Marco Leonec and Julie Morin for the RT-qPCR and Luminex assays, and for the preparation of reagents; Sophie Luccantoni, Céline Mayet, Camille Mabillon for the NHP tissue processing and histology staining; Anne-Sophie Gallouët, Mathilde Galhaut, Cécile Hérate for the help in NHP experiments and tissue processing; Blanche Fert for the CT experiments; Mylinda Barendji, Julien Dinh and Elodie Guyon for the NHP sample processing; Karine Barthelemy for drug concentration analysis; Sylvie Keyser for the transports organization; Frederic Ducancel and Yann Gorin for their help with the logistics and safety management; Isabelle Mangeot for here help with resources management and Brice Targat contributed to data management. The Infectious Disease Models and Innovative Therapies (IDMIT) research infrastructure is supported by the "Programme Investissements d'Avenir", managed by the ANR under reference ANR-11-INBS-0008. The Fondation Bettencourt Schueller and the Region Ile-de-France contributed to the implementation of IDMIT's facilities and imaging technologies. The NHP study received financial support from REACTing, the Fondation pour la Recherche Medicale (FRM; AM-CoV-Path), the "Agence Nationale de Recherche sur le SIDA et les hépatites virales – Maladies infectieuses émergentes" (ANRS-MIE), the ZIKAlliance project which received funding from the European Union's Horizon 2020 Research and Innovation Programme under Grant Agreement N.734548, and the European Infrastructure TRANSVAC2 (730964) for implementation of in vivo imaging technologies an NHP immuno assays. The European Union IMI2 program CARE (101005077) supports the development of the models. The HAE study received financial support from REACTing, the Fondation pour la Recherche Medicale (FRM; AM-CoV-Path) and the Région Auvergne-Rhône-Alpes. The virus stocks used in NHPs were obtained through the EVAg platform (https://www.european-virus-archive.com/), funded by H2020 (653316).

## Author contributions

R.M. contributed to the project conception and design of the study, contributed to animal work, the coordination of the experiments, data analysis, figures design, and the writing of the paper. D.D. contributed to project conception and design of the study, the coordination of the experiments, data analysis, and the writing of the paper. V.C. contributed to project conception and design of the study, contributed to

animal work, data analysis, and figures design. G.L. contributed to data analysis and statistical analysis. C.S. supervised and coordinated the FPV pharmacokinetics analysis, contributed to data analysis and the writing of the paper. P.M. contributed to project conception and design of the study, contributed to animal work. Julien LEMAITRE contributed to clinical follow-up of macaques, to animal work, data analysis, and the writing of the paper. T.N. performed CT scans and quantification, contributed to the data analysis and the writing of the paper. N.K. performed CT scans and acquisition parameter design, and contributed to data analysis. Q.P. contributed to animal work, supervised and coordinated histological analysis, performed tissue lesion scoring, and contributed to the writing of the paper. C.L. contributed to animal work, set up and performed histological staining. N.D.-B. contributed to the animal work and cytokine measurements, analyzed the data, and coordinated IDMIT core activities. B.D. contributed to coordination of animal experiment, contributed to animal work, and acquisition on animal data. C.C. coordinated the imaging facility. R.H.T.F. coordinated the animal core facility, and contributed to study design and data analysis. Francis Relouzat contributed to animal work. A.N. performed RT–qPCR viral quantification and analyzed the data. S.B. performed RT–qPCR viral quantification and analyzed the data. S.v.d.W. provided the viral challenge stock, coordinated the viral load quantification, and analyzed the data. O.T. and A.P. designed the in vitro evaluation of FPV (HAE), performed in vitro work, and analyzed the data. M.R. Calatrava designed the in vitro evaluation of FPV (HAE), supervised and coordinated the work, analyzed the data. D.M. contributed to data analysis and the writing of the paper. P.R. contributed to the project conception and design of the study, the coordination of the experiments, data analysis, and the writing of the paper. X.d.L. contributed to study design, pharmacokinetics/pharmacodynamics analysis, and the writing of the paper. R.L.G. conceived the project, designed the study, coordinated the work, analyzed the data, and wrote the article. J.G. contributed to project conception and study design, to data analysis, the pharmacokinetics/pharmacodynamics study, and wrote the article.

## Competing interests

The funders had no role in study design, data collection, data analysis, data interpretation, or data reporting. The authors D.M., C.S., J.G., and X.d.L. report ongoing collaborations between their institution, INSERM, and Toyama, the manufacturer of Favipiravir. The remaining authors declare no competing interests.

## Additional information

¹Université Paris-Saclay, Inserm, CEA, Center for Immunology of Viral, Auto-immune, Hematological and Bacterial diseases » (IMVA-HB/IDMIT), Fontenay-aux-Roses & Le Kremlin-Bicêtre, France. ²Université de Paris, INSERM, IAME, F-75018 Paris, France. ³Aix-Marseille Univ, APHM, Unité des Virus Emergents (UVE) IRD 190, INSERM 1207, Laboratoire de Pharmacocinétique et Toxicologie, Hôpital La Timone, 13005 Marseille, France. ⁴Unité de Génétique Moléculaire des Virus à ARN, GMVR, Institut Pasteur, UMR CNRS 3569, Université de Paris, Paris, France. ⁵Centre National de Référence des Virus des infections respiratoires (dont la grippe), Institut Pasteur, Paris, France. ⁶CIRI, Centre International de Recherche en Infectiologie, (Team VirPath), Univ Lyon, Inserm, U1111, Université Claude Bernard Lyon 1, CNRS, UMR5308, ENS de Lyon, F-69007 Lyon, France. ⁷Unité des Virus Emergents, UVE: Aix Marseille Univ, IRD 190, INSERM 1207, IHU Méditerranée Infection, 13005 Marseille, France. ⁸VirNext, Université Claude Bernard Lyon 1, Faculté de Médecine Laennec, Lyon, France. ⁹Department of infectious ad tropical diseases, University hopsital, Bordeaux & Inserm 1219/IRD, Bordeaux University, Bordeaux, France. ¹⁰Present address: Virology Unit, Institut Pasteur de Guinée, Conakry, Guinée. ¹¹These authors contributed equally: Romain Marlin, Delphine Desjardins, Vanessa Contreras, Guillaume Lingas, Caroline Solas, Pierre Roques, Jeremie Guedj, Roger Le Grand. ✉e-mail: jeremie.guedj@inserm.fr; roger.le-grand@cea.fr

