## [Peer Review File · Nature Communications]

REVIEWER COMMENTS

Reviewer #1 (Remarks to the Author):

Summary

The authors have conducted a series of preclinical studies using cynomolgus macaques to assess pharmacokinetics and antiviral efficacy of favipiravir (FPV) against Zika virus (ZIKV) and severe acute respiratory syndrome coronavirus 2 (SARS-CoV-2). Collectively, authors treated 42 cynomolgus macaques (20 females and 22 males) with various regimens of FPV or vehicle following infection with either ZIKV or SARS-CoV-2.

They evaluated the pharmacokinetics of FPV in the first experiment (n=4, uninfected), antiviral activity of FPV against ZIKV in the second experiment (n=12) and the efficacy of a range of FPV loading (200 and 250 mg/kg) and maintenance (100, 150 and 180 mg/kg) doses against SARS-CoV-2 in the third experiment (n=30). They monitored endpoints that include: viral load in plasma (ZIKV) and upper respiratory tract (nasopharyngeal and tracheal), bronchoalveolar lavage (BAL) or rectal specimens (SARS-CoV-2).

FPV appears to have rather weak antiviral activity for reducing ZIKV RNA when given as pre-exposure prophylaxis following ZIKV infection. FPV does not appear effective for preventing or reducing viral replication following SARS-CoV-2. Four SARS-CoV-2-infected, then FPV treated animals were euthanized prior to study completion. The authors contend that the rapid clinical deterioration of four animals suggests a potential role of FPV in SARS-CoV-2 disease exacerbation.

Major Concerns

In general, these studies have been well-conducted by a very experienced group of investigators. My overall enthusiasm for this work was diminished by 3 major issues. First, the antiviral activity of FPV against Zika was quite limited at best. Moreover, its utility in ZIKV infection is further diminished by its contraindicated use in pregnancy.

Second, the authors do not make a strong case that the FPV is exacerbating SARS-CoV-2 infection. The data suggests that FPV dosing, combined with ketamine and medetomidine sedation (all 3 drugs are metabolized by the liver), results in the observed pathology. Unfortunately, no data from an FPV “only” arm with matched blood draws (using the same frequency of ketamine and medetomidine sedation) was provided to help interrogate this issue.

Third, the authors have not shown the distribution of animal sex (or animal age) in each study or arm. Males and females in both SARS-CoV-2 and ZIKV infection have dramatically different viral burdens and cytokine profiles. Animal age is also a factor. This information needs to be provided to appropriately analyze each study outcome. Additional minor concerns are noted below.

Statistics

1. Authors should include a statistics section describing the statistical analyses applied in the manuscript Methods section.
2. Please define the provide specific statistics applied lines or the error bars throughout all figures.
3. P value shown in each figure is confusing. Did authors intend to show only significant ones? There are multiple insignificant p values shown in figures 3f & e, 4f&g, S1d, and S5e. Where relevant, please provide exact values for both significant and non-significant P values.
4. Please be consistent to report P values for the representative figures throughout the manuscript. Authors used (P-value = XX), (P=XX), (p-value=XX) or (p=XX).
5. P values shown in figures are different from what was reported in the results section.
 - a. It was reported that P values were 0.052 and 0.19 for Fig. 3e and 3g, respectively in the Results (lines 119-120). However, it was shown as 0.056 and 0.193 in Fig. 3e and f.
 - b. P value was reported as 0.087 in the Results (line 155). However, it was shown as 0.053 in Fig. S35e.
6. Figures 3e & i: Authors performed a spearman correlation test to assess the relationship between peak and total viral burden (AUC) and plasma trough concentration of FPV. Whether this is an appropriate analysis is questionable because the trend toward a concentration dependent FPV effect on the control of viral replication appears to be driven by higher viral load in untreated animals (FPV 0 ug/ml). Among the treated animals, the geometric mean of plasma trough concentration doesn't seem to be correlated with the control of both viral peak and AUC as authors mentioned (lines 119-120). There is a split between <200 ug/ml and >200 ug/ml treated animals.
7. Figures 4c-e: Authors performed a nonparametric Mann-Whitney test to compare values from single treated group to those from untreated control group, which compares the distributions of two unmatched groups as indicated in the figure legend (line 287). There are four groups in the third experiment including one untreated and three treated with escalated concentrations of FPV. A non-parametric one-way ANOVA (Kruskal-Wallis test) following multiple comparisons to adjust P values should be used for the analysis.

8. Figure 4d: How did authors calculate area under the curve (AUC) indicated as Viral load AUC0-7 (Log10 copies.day/mL) on y-axis? Median values that authors mentioned in the manuscript “45.9, 40.4, 48.5 and 49.2 log10 copies day/mL in untreated, 100 mg, 150 mg and 180 mg/kg BID groups, respectively” (lines 148-149) needs clarification.

9. Figure S2. Please provide specific statistics used to compare the differences in the concentration of each cytokine between control and treated arms.

10. Figure S5: Please use a non-parametric one-way ANOVA (Kruskal-Wallis test) following multiple comparisons to adjust P values for the analysis.

11. Figure S9d: It is meaningless to claim “Furthermore, high amount of virus was found in lung tissues of these four animals” in the result (lines 197-198) if authors failed to described viral RNA levels in lung tissues isolated from other infected and treated animals to make comparisons.

Figure Legends

1. Figure Legends titles are duplicated (lines 249-250).

2. Figure 1 legend: FPV dosing regimen described in figure legend (lines 257-259) is different from the schematic shown in Fig. 1. The loading doses (200 or 250 mg/kg) were administered not on day 0 but 3 or 2 days prior to ZIKV or SARS-CoV-2 exposure?

Heatmaps

1. The color scale: how did authors define the range of color scale from the smallest to the largest values? The color for the highest values on the scale does not seem to match with the color used in Heatmaps.

2. Range of the heatmap scale: Authors showed different scales for the same IFN-gamma, IL-2, IL-6, IL-8, CCL3, CCL4 and TNF-alpha between Fig. S2 (ZIKV) and S7 (SARS-CoV-2). Authors claimed that the sustained high concentration of cytokine levels shown in FPV treated animals is casually related to FPV administration. However, it seems cytokine levels between control and treated animals are different following different virus infections.

Although it may not be necessary to directly compare cytokine levels in ZIKV infected animals to those levels in SARS-CoV-2 infected animals following infection, authors at least should show those figures using the same scales.

3. How did author measure the concentration of each cytokine? Using ELISA or Luminex multiplex assay? Please describe in the Methods.

Methods

1. A total of 42 cynomolgus monkeys (20 females and 22 males) were used in the study. However, authors did not clarify how they distributed females and males in each study and study arms. For both ZIKV and SARS-CoV2 infections there are multiple parameters (including viral load and cytokine profiles) that are skewed by sex.

2. Please use the same identification of the SARS-CoV-2 strain throughout the manuscript. It was indicated as (WHA-D164G, BetaCoV/France/IDF/0372/2020), but (hCoV-19/France/ IDF0372/2020 SARS-CoV-2 strain; ; GISAID EpiCoV platform under accession number EPI_ISL_406596) in the Methods.

References

1. References# 7 and 8 are duplicated (lines 606-610).

Reviewer #2 (Remarks to the Author):

In this manuscript, Marlin et al., evaluated the efficacy of the influenza drug favipiravir against ZIKV and SARS-CoV-2 in NHP model. The drug showed some antiviral efficacy against ZIKV in that model but not against SARS-CoV-2. Moreover, 4 animals that were treated with the drug and infected with SARS-CoV-2 showed rapid clinical deterioration. Overall, the manuscript is well written and presents very important data especially the results against SARS-CoV-2 as favipiravir is involved in several clinical trials for COVID19. I have only minor questions and comments:

1. the design of experiments need a bit clarification:

-it is not clear for me why the compound was not administered orally like the case in humans and why the loading dose was by IV and not SC like the rest of dosing.

- Why was the challenge started 2 days after first treatment for SARS-CoV-2 while it was 3 days for ZIKV?

2. For ZIKV: in line 214 in discussion it is mentioned that favipiravir can penetrate sexual organs and BBB, have you checked for viral loads in sexual organs and nervous system of the ZIKV infected animals?

3. line: 116, the values for viral shedding are switched (i.e. 6.5 log₁₀ should be for the untreated animals not the treated ones) so please correct.

4. what are the possible reasons for the clinical deterioration observed in the SARS-CoV-2 animals? is there any reported data of immune effects associated with Favipiravir?

RESPONSES TO REVIEWERS' COMMENT

Dear Editor,

We would like to thank you and the two anonymous reviewers for their interest and thorough reviews, which greatly improved the manuscript. Please find below our rebuttal letter.

Reviewer #1 (Remarks to the Author):

We thank Reviewer #1 for his positive and constructive comments on our manuscript.

- 1. The reviewer commented that,** *“First, the antiviral activity of FPV against Zika was quite limited at best. Moreover, its utility in ZIKV infection is further diminished by its contraindicated use in pregnancy.”*

Authors response: The reviewer is correct about the limitations of FPV. We have extended our discussion on the potential role of FPV against Zika in the discussion: *“The effects of favipiravir on Zika viremia in macaques unambiguously demonstrate for the first time in vivo antiviral activity of favipiravir. As mentioned for other viral diseases, favipiravir can penetrate the sexual compartments and cross the blood brain barrier, facilitating the purge of the reservoirs of the virus. In addition to mono-therapy, favipiravir may represent a good candidate for combined therapies with antiviral drugs such as galidesivir, which was also reported highly active in the NHP model. Although the probable teratogenicity of favipiravir prevents its use in pregnant women, its ease of storage and its oral administration make it particularly relevant as a first line of protection to administer to suspect or contact cases.”*

For the reviewer information, we add data regarding the ability of favipiravir to penetrate the sexual compartments (data from 6 animals infected with Zika virus).

Seminal plasma were collected at 2 and 10 dpi. Median value is indicated by horizontal bar.

- 2. The reviewer commented that,** *“Second, the authors do not make a strong case that the FPV is exacerbating SARS-CoV-2 infection. The data suggests that FPV dosing, combined with ketamine and medetomidine sedation (all 3 drugs are metabolized by*

the liver), results in the observed pathology. Unfortunately, no data from an FPV “only” arm with matched blood draws (using the same frequency of ketamine and medetomidine sedation) was provided to help interrogate this issue”.

Authors response: We have repeatedly tested the use of FPV in combination sedation in different experimental settings and exacerbation of disease was only observed in SARS-CoV-2 exposed animals. During the PK study (FPV only arm) performed in non-infected animals, sedation was combined with treatment multiple times during the 15 days of follow up.

In the ZIKV study, frequency of sedation was similar to SARS-CoV-2 study and do not result in exacerbation of ZIKV infection, nor exacerbation of signs of liver damage. In addition, exacerbation of disease we reported in FPV treated and SARS-CoV-2 infected animals, was associated with an important increase of inflammatory response and clear signs of severe pneumonia typical of viral infection induced severe acute respiratory syndrome. Of note, the control arm of the study include macaques exposed to SARS-CoV-2 with identical sedation procedure and frequency. No exacerbated disease was reported in this group, confirming our results published in several therapeutic and vaccine studies we performed in this specie, with same challenge dose and virus strain and similar sedation frequency to what we report here.

In addition we previously conducted study of favipiravir administered IV with similar frequency of sedation (<https://doi.org/10.1128/AAC.01305-16>). Altogether, these data strongly suggest that the combination of SARS-CoV-2 infection and FPV treatment was responsible to exacerbated pathology.

Regarding the metabolism, favipiravir is extensively metabolized through aldehyde oxydase (xanthine oxydase to a lower extent) and does not interact with both ketamine (CYP3A4, CYP2B6 and CYP2C9 substrate) and medetomidine (UGT1A4/2B10 and CYP2A6 substrate). Favipiravir inhibits CYP2C8 that is not involved in their metabolism either.

We have added the following sentences to the discussion: *“Our results showed also an exacerbation of SARS-CoV-2 disease in four animals treated with favipiravir, with an impact on liver and kidney functions. Importantly this effect had not been seen in previous experiments involving favipiravir and repeated anesthesia nor in infected untreated animals, suggesting that this effect was causally related to the dual effects of favipiravir administration and SARS-CoV-2 infection. “*

- 3. The reviewer also noted that,** *“Third, the authors have not shown the distribution of animal sex (or animal age) in each study or arm. Males and females in both SARS-CoV-2 and ZIKV infection have dramatically different viral burdens and cytokine profiles. Animal age is also a factor. This information needs to be provided to appropriately analyze each study outcome”*

Authors response: We are now providing these information in **supplementary table I** that included sex and age of each NHP. Moreover, we are also providing in **supplementary fig. 10** viral kinetics and cytokine profile in our historical NHP infected with SARS-CoV-2 used as controls of therapeutic and vaccine trials. This supplementary figure shows that in our animal model, SARS-CoV-2 infection is similar in male and female cynomolgus macaques.

Statistics

1. Authors should include a statistics section describing the statistical analyses applied in the manuscript Methods section.

Authors response: We do agree with the reviewer that this section is missing. We have now added a statistic section (lines 549-557).

2. Please define the provide specific statistics applied lines or the error bars throughout all figures.

Authors response: We do agree with the reviewer that this point is missing. We have now added this information in all figure legends.

3. P value shown in each figure is confusing. Did authors intend to show only significant ones? There are multiple insignificant p values shown in figures 3f & e, 4f&g, S1d, and S5e. Where relevant, please provide exact values for both significant and non-significant P values.

Authors response: We do agree with the reviewer that this point was confusing. We have modified the figures and removed non-significant P values, only significant (<0.05) or trend (<0.08) values are indicated.

4. Please be consistent to report P values for the representative figures throughout the manuscript. Authors used (P-value = XX), (P=XX), (p-value=XX) or (p=XX).

Authors response: We have homogenized the manuscript and used p=XX.

5. P values shown in figures are different from what was reported in the results section.
a. It was reported that P values were 0.052 and 0.19 for Fig. 3e and 3g, respectively in the Results (lines 119-120). However, it was shown as 0.056 and 0.193 in Fig. 3e and f.

Authors response: Thanks for pointing out these typos. We have modified the text accordingly.

b. P value was reported as 0.087 in the Results (line 155). However, it was shown as 0.053 in Fig. S35e.

Authors response: We do agree with the reviewer and we have corrected the error. Reported p values in text was correct.

6. Figures 3e & i: Authors performed a spearman correlation test to assess the relationship between peak and total viral burden (AUC) and plasma trough concentration of FPV. Whether this is an appropriate analysis is questionable because the trend toward a concentration dependent FPV effect on the control of viral replication appears to be driven by higher viral load in untreated animals (FPV 0 ug/ml). Among the treated animals, the geometric mean of plasma trough concentration doesn't seem to be correlated with the control of both viral peak and AUC as authors mentioned (lines 119-120). There is a split between <200 ug/ml and >200 ug/ml treated animals.

Authors response: We respectfully disagree with the reviewer on this aspect. One of the strength of our analysis is to rely on drug exposure, which increases the capability to identify a treatment effect. As the reviewer points out, if favipiravir has a genuine antiviral activity, then higher exposure should be associated with a larger antiviral effect. Such signal is

observed for ZIKV peak but not for ZIKV AUC (Fig 3e-f). We nonetheless agree with the reviewer that these results suggest that the exposure/effect relationship is probably nonlinear. We have added the following comment in the results

“Drug concentration showed a trend towards an effect on peak viral load and on AUC viral load ($p=0.056$ and $p=0.074$, respectively, Fig 3e-f), suggesting that high concentrations could be associated with a reduction of viral load, with a nonlinear relationship. “

7. *Figures 4c-e: Authors performed a nonparametric Mann-Whitney test to compare values from single treated group to those from untreated control group, which compares the distributions of two unmatched groups as indicated in the figure legend (line 287). There are four groups in the third experiment including one untreated and three treated with escalated concentrations of FPV. A non-parametric one-way ANOVA (Kruskal-Wallis test) following multiple comparisons to adjust P values should be used for the analysis.*

Authors response: We do agree with the reviewer and we have now performed Kruskal-Wallis test following Dunn’s multiple comparisons to compared the four groups in SARS-CoV-2 study. Figure legends, p-value and method section was updated following this change. Interpretation of results was not changed since statistic difference previously noticed, remained statistically significant using Kruskal-Wallis test.

8. *Figure 4d: How did authors calculate area under the curve (AUC) indicated as Viral load AUC0-7 (Log10 copies.day/mL) on y-axis? Median values that authors mentioned in the manuscript “45.9, 40.4, 48.5 and 49.2 log10 copies day/mL in untreated, 100 mg, 150 mg and 180 mg/kg BID groups, respectively” (lines 148-149) needs clarification.*

Authors response: We thank the reviewer for point out the discrepancy in the calculation of AUC, that was calculated either as the logarithm of the AUC of the viral load, or as the AUC of the log of the viral load. We now only present one metric, the logarithm of the AUC of the viral load and we have clarified this aspect in the methods. All results and figures have been updated accordingly.

9. *Figure S2. Please provide specific statistics used to compare the differences in the concentration of each cytokine between control and treated arms.*

Authors response: A nonparametric Mann-Whitney test was performed to compare cytokine concentration between experimental groups. However, there was an error in previous figure S2, since difference was noticed 5 dpi but not 3 dpi. Error was corrected in the new Figure S2.

10. *Figure S5: Please use a non-parametric one-way ANOVA (Kruskal-Wallis test) following multiple comparisons to adjust P values for the analysis.*

Authors response: We do agree with the reviewer and we have now performed Kruskal-Wallis test following Dunn’s multiple comparisons to compared the four groups in SARS-CoV-2 study. Figure legends, p-value and method section was updated following this change

11. *Figure S9d: It is meaningless to claim “Furthermore, high amount of virus was found in lung tissues of these four animals” in the result (lines 197-198) if authors failed to described viral*

RNA levels in lung tissues isolated from other infected and treated animals to make comparisons.

Authors response: We do agree with the reviewer that this point needs to be clarified. Viral RNA levels in lung tissues from other animals of the study were illustrated in fig S5g. For greater clarity, these data was also now presented in fig. S9d to make comparisons with the data of the 4 animals with exacerbated disease.

Figure Legends

1. *Figure Legends titles are duplicated (lines 249-250).*

Authors response: Duplicated title is now deleted.

2. *Figure 1 legend: FPV dosing regimen described in figure legend (lines 257-259) is different from the schematic shown in Fig. 1. The loading doses (200 or 250 mg/kg) were administered not on day 0 but 3 or 2 days prior to ZIKV or SARS-CoV-2 exposure?*

Authors response: We have corrected the error in the figure's legend.

Heatmaps

1. *The color scale: how did authors define the range of color scale from the smallest to the largest values? The color for the highest values on the scale does not seem to match with the color used in Heatmaps.*

Authors response: We do agree with the reviewer that this point needs to be clarified. For each cytokine, range of color scale are the same between fig 3g (ZIKV) and Fig 4i (SARS-CoV-2) in order to highlight the peak and the maintenance of high cytokine level during viral infection in combination or not with FPV treatment. For specific time point and animal, concentration value are above the highest values of the scale and are indicated in specific color Red. For clarity, we have now indicated this color in the range scale. Moreover, for these specific conditions, concentration values are indicated directly within the heat-maps.

2. *Range of the heatmap scale: Authors showed different scales for the same IFN-gamma, IL-2, IL-6, IL-8, CCL3, CCL4 and TNF-alpha between Fig. S2 (ZIKV) and S7 (SARS-CoV-2). Authors claimed that the sustained high concentration of cytokine levels shown in FPV treated animals is casually related to FPV administration. However, it seems cytokine levels between control and treated animals are different following different virus infections.*

Although it may not be necessary to directly compare cytokine levels in ZIKV infected animals to those levels in SARS-CoV-2 infected animals following infection, authors at least should show those figures using the same scales.

Authors response: We do agree with the reviewer and we have homogenized the heat maps and the scales. Now for each cytokine, concentrations scale is similar between ZIKV (Fig. S2) and SARS-CoV-2 data (Fig. S7)

3. *How did author measure the concentration of each cytokine? Using ELISA or Luminex multiplex assay? Please describe in the Methods.*

Authors response: Cytokines and chemokines were measured using Luminex multiplex assay. This information is now added in method section (lines 458-460).

Methods

1. A total of 42 cynomolgus monkeys (20 females and 22 males) were used in the study. However, authors did not clarify how they distributed females and males in each study and study arms. For both ZIKV and SARS-CoV2 infections there are multiple parameters (including viral load and cytokine profiles) that are skewed by sex.

Authors response: We are now providing these information in **supplementary table I** that included sex and age of each NHP.

2. Please use the same identification of the SARS-CoV-2 strain throughout the manuscript. It was indicated as (WHA-D164G, BetaCoV/France/IDF/0372/2020), but (hCoV-19/France/IDF0372/2020 SARS-CoV-2 strain; ; GISAID EpiCoV platform under accession number EPI_ISL_406596) in the Methods.

Authors response: We have homogenized the manuscript and SARS-CoV-2 strain is indicated as hCoV-19/France/IDF0372/2020.

References

1. References# 7 and 8 are duplicated (lines 606-610).

Authors response: Duplicated reference is now deleted.

Reviewer #2 (Remarks to the Author):

We thank Reviewer #2 who considers that, *“the manuscript is well written and presents very important data especially the results against SARS-CoV-2 as favipiravir is involved in several clinical trials for COVID19.”*

We hope that our responses will be favorably consider by the reviewer.

1. The reviewer commented that, *“the design of experiments need a bit clarification”*:

-it is not clear for me why the compound was not administered orally like the case in humans and why the loading dose was by IV and not SC like the rest of dosing.

Authors response: Favipiravir was previously tested in filovirus infection, where oral and IV route were used. Bixler et al. (<https://doi.org/10.1016/j.antiviral.2017.12.021>) showed that oral route did not confer a survival benefit in filovirus-infected NHPs, whereas survival was significantly improved when FPV was administrated by IV route. Authors suggested also that oral delivery could reduce systemic exposure. For these reasons, we did choose the SC route in order to obtain the maximal exposure of animals to the molecule, as confirmed by our PK study, with an experimental design given the best chances to unravel an antiviral effect. Also, loading dose was delivery by IV in order to reach the target trough concentration as fast as possible. Furthermore, the SC route limited the need for repeated sedation of NHP, this route was thus preferred for the maintenance dose.

- Why was the challenge started 2 days after first treatment for SARS-CoV-2 while it was 3 days for ZIKV?

Authors response: First modeling data obtained during the PK study showed that 3 days were necessary to stabilize the systemic FPV concentration and reach a trough concentration close to the *in vitro* efficient concentration. Thus we performed ZIKV infection, three days after FPV treatment initiation. However, FPV dosage and modeling data obtained from ZIKV study showed that in fact C_{trough} was reached in two days.

2. The reviewer commented that, *“For ZIKV: in line 214 in discussion it is mentioned that favipiravir can penetrate sexual organs and BBB, have you checked for viral loads in sexual organs and nervous system of the ZIKV infected animals?”*

Authors response: We indeed sought ZIKV and FPV in seminal plasma during the study. Thus, we were able to detected ZIKV in one NHP (1374 viral copies/mL) at 10 dpi. Moreover, FPV was quantified in seminal plasma at 2 and 10 dpi. We measured a median concentration of 24.75 $\mu\text{g/mL}$ (see figure below). Unfortunately, we did not collect cerebrospinal fluid during the ZIKV study.

Seminal plasma FPV concentration of 6 ZIKV infected NHPs. Seminal plasma were collected at 2 and 10 dpi. Median value is indicated by horizontal bar. (*for reviewer only*)

3. line: 116, the values for viral shedding are switched (i.e. 6.5 log₁₀ should be for the untreated animals not the treated ones) so please correct.

Authors response: We have corrected the error in the manuscript.

4. what are the possible reasons for the clinical deterioration observed in the SARS-CoV-2 animals? is there any reported data of immune effects associated with Favipiravir?

Authors response: Altogether, our data showed the clinical deterioration observed in the 4 SARS-CoV-2 animals was due to multiple organ dysfunction.

However, to our knowledge, effect of FPV on immune cells was not described so far. Lee et al. (<https://doi.org/10.1073/pnas.0631696100>) showed that guanine nucleoside analogs could activate TLR-7 and thus, indirect effect of FPV on immune cells could not be excluded. However, severe pneumonia we confirmed in these animals by CT scan suggest strong interaction between viral dynamics and Favipiravir resulting in the exacerbated disease.

REVIEWERS' COMMENTS

Reviewer #1 (Remarks to the Author):

Overall, the authors responded to most of the major concerns. There should be added discussion regarding SARS-CoV-2 infection of the liver and kidney of which there is substantial literature. Many of the citations still need updating. No additional concerns beyond that.

RESPONSES TO REVIEWERS' COMMENT

Reviewer #1 (Remarks to the Author):

We thank Reviewer #1 for his positive comment.

- 1. The reviewer commented that, “There should be added discussion regarding SARS-CoV-2 infection of the liver and kidney of which there is substantial literature.”**

Authors response: We have extended our discussion on the SARS-CoV-2 infection of liver and kidney in the discussion section (lines 236-244):

“SARS-CoV-2 infection results in several extrapulmonary manifestations, including kidney and liver injury³⁰. Angiotensin-converting enzyme 2 (ACE2), the entry receptor for SARS-CoV-2, is expressed in the liver and kidney³¹⁻³³ but viral replication in these organs remains controversial^{34,35}. Both kidney and liver injury are likely multifactorial involving direct effects of the virus with inflammation and tissue damages, but also indirect effects resulting from systemic inflammation, dysregulated immune responses, endothelial dysfunction, and impaired organ crosstalk^{32,35,36}. Elevation of liver enzymes have been reported during SARS-CoV-2 infection, however liver injury may be a reflection of a severe form of the disease³². Hepatic injury associated with COVID-19 seems due to systemic inflammation and multi-organ dysfunction³⁶.”